# Is Integer Arithmetic Enough for Deep Learning Training?

**Alireza Ghaffari**[1]    **Marzieh S. Tahaei**[1]    **Mohammadreza Tayaranian**[1]
**Masoud Asgharian**[2]    **Vahid Partovi Nia**[1]
[1] Huawei Noah's Ark Lab, Montreal Research Center
[2] Department of Mathematics and Statistics, McGill University
{alireza.ghaffari, marzieh.tahaei, mohammadreza.tayaranian}@huawei.com
vahid.partovinia@huawei.com    masoud.asgharian2@mcgill.ca

## Abstract

The ever-increasing computational complexity of deep learning models makes their training and deployment difficult on various cloud and edge platforms. Replacing floating-point arithmetic with low-bit integer arithmetic is a promising approach to save energy, memory footprint, and latency of deep learning models. As such, quantization has attracted the attention of researchers in recent years. However, using integer numbers to form a fully functional *integer* training pipeline including forward pass, back-propagation, and stochastic gradient descent is not studied in detail. Our empirical and mathematical results reveal that *integer arithmetic* seems to be enough to train deep learning models. Unlike recent proposals, instead of quantization, we directly switch the number representation of computations. Our novel training method forms a fully integer training pipeline that does not change the trajectory of the loss and accuracy compared to floating-point, nor does it need any special hyper-parameter tuning, distribution adjustment, or gradient clipping. Our experimental results show that our proposed method is effective in a wide variety of tasks such as classification (including vision transformers), object detection, and semantic segmentation.

## 1   Introduction

Recently, deep learning models have evolved to deliver acceptable performance in many areas such as computer vision, natural language processing, and speech recognition. However, the number of parameters and computational complexity of most deep learning applications have increased significantly. This ever-increasing computational complexity makes the deployment of deep neural networks harder on edge devices. Furthermore, the energy required to train a deep learning model increases proportionately with computational complexity. Thus, the training cost of these large and complex models also increases significantly on the high-end cloud servers. Although, quantization techniques are commonly used to accelerate *inference* of deep learning models, here we target a fully integer training pipeline *(i.e. forward propagation, back-propagation and SGD)*. Unlike recent proposals in quantized back-propagation, we directly change the number representation of floating-point values. Furthermore, we show empirically and theoretically that our proposed integer training methodology is effective in training deep learning models with *integer-only arithmetic*.

Despite the fact that accelerated training using integer back-propagation seems intriguing, there are some challenges associated with integer gradient computation. For example, (i) gradients must be scaled correctly in order to be adapted to the limited dynamic range of the integer number (e.g. `int8`: $[-128, 127]$). (ii) The numerical error of the gradient must be ***unbiased*** in order to preserve the convergence trajectory of the training algorithm. A small numerical error can be accumulated

36th Conference on Neural Information Processing Systems (NeurIPS 2022).

through the course of training and change the convergence behaviour. (iii) The integer training must be *distribution independent*, i.e. the training method should not depend on the distribution of the gradients, weights, or training data.

We propose a novel integer training method designed to address aforementioned challenges (i), (ii) and (iii) simultaneously. Having a linear dynamic fixed-point mapping coupled with a non-linear inverse mapping is the key to full integer training.

To this end, we make the following contributions:

- A hardware-friendly integer training method is proposed based on extracting the maximum floating-point exponent of the tensors as the *scale*. We propose linear fixed-point mapping of tensors, while the corresponding inverse mapping for floating-point is non-linear. Our proposed method addresses all the previously mentioned challenges: (i) it computes the scales dynamically, and moreover the scale does not need to be adjusted in the course of training; (ii) it provides an unbiased estimation of the gradients and consequently, its convergence trajectory closely follows the floating-point version; (iii) our proposed representation mapping does not depend on the distribution of the training data, weights, or gradients.

- We study the optimality gap of our proposed integer training algorithm and show it is analogous to its floating-point counterpart in the course of training (**Theorem 1**). Our analysis of stochastic gradient descent with our proposed fixed-point gradients shows the original floating-point optimality gap is only shifted by a negligible amount (**Remark 3**).

- Our proposed method effectively performs all operations required in modern neural networks using *integer-only arithmetic*. For instance, the computation of linear layer, convolutional layer, batch-norm and layer-norm, residual connections and also stochastic gradient descent (including gradients, momentum, weight decay, and weight update) are all performed in integer arithmetic. To the best of our knowledge, this is the first time that *back-propagation* of a batch-norm, and the computation of stochastic gradient descent (SGD) is performed in integer arithmetic with negligible loss in the accuracy for large datasets such as ImageNet.

The rest of this paper is structured as follows. Section 2 reviews some previous works in the field of quantized back-propagation and quantifies the similarities and differences with our *representation mapping* method. Section 3 discusses our integer training methodology in detail. Theoretical aspects of our integer training method on SGD are studied in Section 4. Experimental results supporting our integer training methodology and convergence theory are presented in Section 5.

## 2 Related works

Banner et al. [1], proposed a bifurcated back-propagation method where the gradient of weights are in floating-point format and input gradients are in 8-bit fixed-point format. Moreover, they proposed a range-based batch-norm which is more tolerant to quantization noise. The main difference of our proposed algorithm with Banner et al. [1] is that our integer training method does not demand bifurcation. In other words, our gradients with respect to weights and inputs are represented and computed using `int8` format. Furthermore, in our method, the back-propagation of batch-norm, as well as stochastic gradient descent (SGD) algorithm, are also represented and computed using integer values. Zhang et al. [2] have proposed a layer-wise "precision-adaptive" quantization method that does not affect the distribution of the data. In their method, the quantization error is measured and the quantization scale is adjusted over the course of training accordingly. Similarly, in Zhao et al. [3] a distribution adaptive quantization method is proposed that takes into account the distribution of gradients in the channel dimension. In addition, they introduced a gradient clipping strategy in order to normalize the magnitude of the gradients in the back-propagation. The scale of the quantization is adapted to the distribution of the gradients and adjusted iteratively over the course of training. Unlike the distribution adaptive methods such as [2, 3], our proposed integer training method does not depend on the distribution of training data, weights, and gradients. Thus, there is no need to adjust the scale iteratively in the course of training. In Zhu et al. [4], a "direction sensitive gradient clipping method" is proposed to avoid inappropriate updates in the back-propagation. Moreover, they proposed a learning rate scaling that tackles the problem of unbiased gradient error accumulation. In a similar work, Sakr and Shanbhag [5] proposed a precision assignment methodology to improve the convergence of the quantized back-propagation. This precision assignment is based on some criterion

on internal accumulator noise, quantization noise of backward and forward propagation, and gradient clipping. Unlike [4, 5], our proposed method enjoys having the advantage of *unbiased gradients*, hence the convergence trajectory closely follows the floating-point counterpart. Thus, our method does not need any specific gradient correction, gradient clipping strategy, and learning rate corrections as opposed to [4, 5]. Jin et al. [6] proposed a unified fixed-point and parameterized clipping activation to achieve high accuracy. Furthermore, they proposed a method that directly trains the fractional length (i.e. scale) of the fixed-point quantization. Additionally, they use a double forward batch-norm fusion to determine the scaling factor of the batch-norm layer in the forward propagation, while the back-propagation of batch-norm remains floating-point. In contrast, our proposed integer training method is capable of implementing integer variant of the batch-norm layer in forward propagation without double fusion as opposed to [6]. We also perform batch-norm's back-propagation in integer arithmetic which was ignored in the previous works. Note that back-propagation of batch-norm and layer-norm is sensitive to quantization, as such, naive quantization leads to training divergence. Additionally, we perform stochastic gradient descent (SGD) computation including weight update, weight decay, and momentum in integer arithmetic with no significant loss of accuracy. Wang et al. [7] explored an integer-arithmetic implementation of some preliminary CNN architectures such as AlexNet, however they experienced considerable accuracy drop for ImageNet dataset. Moreover, more complicated architectures such as ResNet and transformers are not considered. In our proposed method, we used dynamic fixed-point number format [8] as the primary number format for integer training. This specific fixed-point number format (also known as block floating-point format) has been used previously to quantize *inference* of deep learning models in Courbariaux et al. [8], Drumond et al. [9], and Rouhani et al. [10].

## 3 Methodology

Common quantization methods, which use division and clipping techniques (refer to Appendix A.6), are inefficient for back-propagation. Therefore, we decided to go one step deeper and change the number format directly. Here, we propose a hardware-friendly *number representation mapping* using the dynamic fixed-point number format where the scale is defined *per tensor*. In this approach, each tensor can be represented by its `int8` version while it is multiplied by a shared scale, as opposed to other methods that allow multiple shared scales for different partitions of the tensor (see Rouhani et al. [10, Figure 4]). One shared scale per tensor makes the computations easier in the computing hardware (e.g. CPU or GPU). However, the training algorithm diverges if the representation mapping is not executed properly. To tackle this problem, we suggest two subtle changes; (i) we propose to perform fixed-point mapping of a tensors in a ***linear*** fashion while the inverse mapping is ***non-linear*** as explained in Sections 3.1 and 3.2. This is the key to success of our algorithm since a linear fixed-point mapping allows monotonic conversion of floating-point format to fixed-point, while a non-linear inverse mapping allows preserving information. (ii) We suggest to use stochastic rounding in the back-propagation in a way that preserves the expected value of the tensors as well as their vital statistics, such as the mean. Stochastic rounding in conjunction with representation mapping is crucial to keep the integer training loss trajectory close to its floating-point counterpart.

### 3.1 Linear fixed-point mapping

Our proposed integer training method is based on manipulating the floating-point number format. This is a very simple and effective way of converting floating-point values to fixed-point values as opposed to other commonly used methods that involve division operation. Our method essentially uses *shift* and *round* operations to convert floating-point to fixed-point format, see Figure 1(a).

In this method, a tensor comprising $n$ floating-point numbers $(f_1, f_2, ..., f_n)$ is converted to a fixed-point tensor. First, as shown in Figure 1(a), sign, exponent, and mantissa of each floating-point number are extracted using the *unpack to integer* function. For instance, $f_1$ is unpacked to $s_1, e_1, m_1$ where $(s, e, m)$ is used to denote (*sign*, *exponent*, *mantissa*). We simply find the *maximum* element of $e_1, e_2, \ldots e_n$ i.e. $e_{\max} = \max(e_1, e_2, \ldots e_n)$ to extract the shared scale of the tensor. Subsequently, the original mantissas are shifted to the right according to the difference of their exponent and $e_{\max}$, to get the individual mantissas. For instance, $m_1$ is shifted to right by $e_{\max} - e_1$ which is denoted by $m_1 >> (e_{\max} - e_1)$. By shifting mantissas to right, we intentionally push the small values to the *sub-normal* region, where the most significant bit of mantissa is not 1 in binary format i.e. $(1)_2$, see Zuras et al. [11]. Pushing the mantissas to the sub-normal region is performed to align their exponents

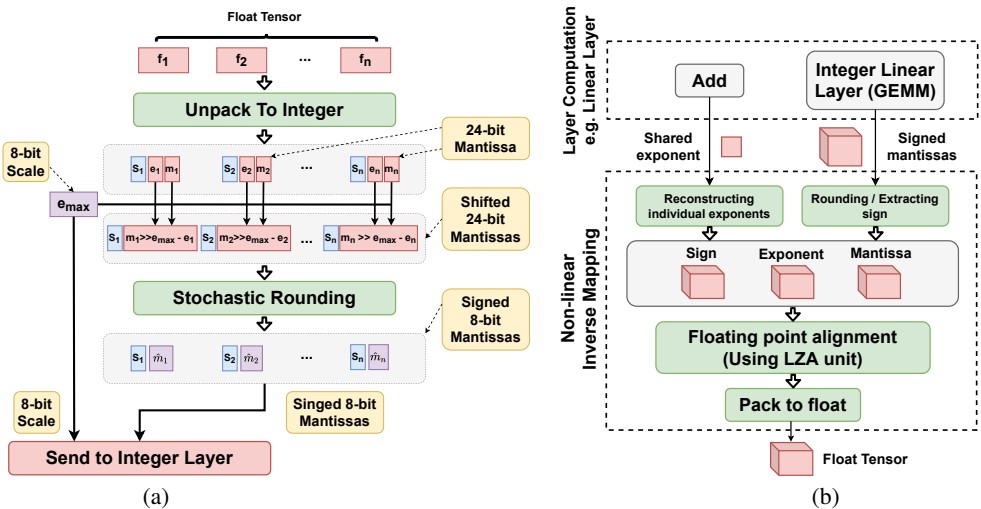

Figure 1: (a) Linear fixed-point mapping, (b) Non-linear inverse mapping.

with $e_{\max}$ and this unifies the scale for fixed-point values in a tensor. In the next step, to create `int8` mantissa, the 24-bit single floating-point mantissas (i.e. 23-bit mantissa + 1 implicit hidden bit) are further rounded to 7-bit mantissas to construct signed `int8` values i.e 7-bit unsigned integer and 1 sign bit. As an example, let us assume the shifted mantissa is $(0.01011001010101010100000)_2$, then it is going to be randomly rounded to either $(0.010110)_2$ or $(0.010111)_2$ based on a probability (Refer to A.1 for details).

## 3.2 Non-linear inverse mapping

Our proposed fixed-point mapping method is performed by pushing mantissa values to the sub-normal region and rounding them stochastically. Hence, the inverse mapping must be performed using a floating-point alignment module that normalizes the mantissas. A *normalized* floating-point mantissa is required to start with the most significant bit $(1)_2$. For instance, an alignment module converts $2^{127} \times (0.0101)_2$ to $2^{127-2} \times (1.0100)_2$. The alignment module is a very well-known logic circuit that is available in the commodity hardware using Leading Zero Anticipator (LZA) block [12]. Coupling a linear fixed-point mapping with a non-linear inverse mapping in this way keeps the number format close to floating-point. Moreover, combining them with stochastic rounding, results in having unbiased fixed-point variant that forces the trajectory of the integer training closely follow the floating-point counterpart.

Figure 1(b) shows the inverse mapping unit. This unit receives a tensor of mantissas and a single value of shared exponent computed by an integer layer. Then a tensor of exponents is reconstructed by repeating the value of the shared exponent. The size of this tensor is equal to the mantissa tensor's size. Moreover, the mantissa tensor is rounded and its sign is extracted. Note that the rounding is used here because the output of the previous layer might have some excessive mantissa bits; this normally happens in matrix multiplication and convolution operations. Then the sign, exponent, and mantissa tensors are sent to an alignment unit that shifts the mantissa and adjust the exponent in order to normalize the floating-point number format [11]. Finally, the result is packed and carried out to the next layer.

## 3.3 Integer layer computations

When the fixed-point mapping of floating-point values is completed, the fixed-point values are used to perform the *integer-only* layer computations. For instance, let us consider a linear layer which consists of a General Matrix Multiplication (GEMM) operation, but, the idea can be generalized to other types of layers. Figure 2 demonstrates the layer-wise integer computations of a linear layer where the shared exponents and integer mantissas are treated separately. As shown in the Figure 2, in order to perform an integer-only GEMM operation, we need to multiply scales ($2^{e_{\max 1}} \times 2^{e_{\max 2}}$). This multiplication performs an integer addition operation on the exponents ($e_{\max 1} + e_{\max 2}$). Furthermore, the integer mantissas are sent to an integer GEMM module to compute the output mantissa tensor.

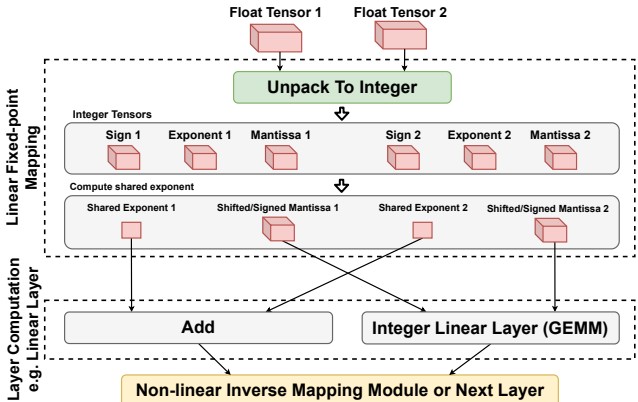

Figure 2: A fully integer linear layer.

Also note that in our implementation, when the mantissa tensor is in `int8` format, multiplication is in `int16` format and accumulation is in `int32` format.

## 3.4 Understanding the representation mapping

Here we provide an intuition of how our proposed representation mapping approach works. Let us denote $A$ as a tensor and $\hat{A}$ as its fixed-point version in the way that is introduced earlier. In addition, random variables $A_i$ and $\hat{A}_i$ are $i^{\text{th}}$ element of those tensors. Also note that $A_i$ and $\hat{A}_i$ are different *representations* of the same real number, but one is in floating-point format and the other is in dynamic fixed-point format. Thus, one can relate $A_i$ and $\hat{A}_i$ with a random error term $\delta_i$ as $\hat{A}_i = A_i + \delta_i$. Since in our training method we used stochastic rounding [13], $\mathbb{E}\{\hat{A}_i\} = A_i$, or equivalently $\mathbb{E}\{\delta_i\} = 0$ (see Appendix A.1). In other words, the fixed-point value is on the average equal to the floating-point value. Note that here we consider single precision floating-point values as a surrogate of real values.

**Linear and convolutional layers:** For these two types of layers, both forward and backward propagation computations are based on inner products. Thus, it is easy to see that our proposed fixed-point inner product $\hat{C}_{ij} = \sum_k \hat{A}_{ik} \hat{B}_{kj}$ is an unbiased estimator of the floating-point inner product

$$\mathbb{E}\{\hat{C}_{ij}\} = \mathbb{E}\left\{\sum_k \hat{A}_{ik}\hat{B}_{kj}\right\} = \mathbb{E}\left\{\sum_k (A_{ik} + \delta_{ik}^A)(B_{kj} + \delta_{kj}^B)\right\} = \sum_k A_{ik}B_{kj} = C_{ij}. \quad (1)$$

**Residual connections:** A residual connection involves element-wise addition of two tensors. Each fixed-point element $\hat{C}_{ij} = \hat{A}_{ij} + \hat{B}_{ij}$ is an unbiased estimator of its floating-point version

$$\mathbb{E}\{\hat{C}_{ij}\} = \mathbb{E}\{\hat{A}_{ij} + \hat{B}_{ij}\} = \mathbb{E}\{(A_{ij} + \delta_{ij}^A) + (B_{ij} + \delta_{ij}^B)\} = A_{ij} + B_{ij} = C_{ij}. \quad (2)$$

**Batch-norm:** A batch-norm layer is defined as

$$\hat{\omega}\frac{\hat{A} - \hat{\mu}}{\sqrt{\hat{\sigma}^2 + \epsilon}} + \hat{\beta}. \quad (3)$$

For this type of layer, it is important to compute signal statistics such as mean $\hat{\mu}$ and variance $\hat{\sigma}^2$ correctly in integer arithmetic. Thus, the fixed-point mean $\hat{\mu}$ is also an unbiased estimator of the true mean $\mu$

$$\mathbb{E}\{\hat{\mu}\} = \mathbb{E}\left\{\frac{\sum_{i=1}^N \hat{A}_i}{N}\right\} = \frac{1}{N}\mathbb{E}\left\{\sum_{i=1}^N (A_i + \delta_i)\right\} = \mu. \quad (4)$$

Then the following derivation holds for fixed-point variance $\hat{\sigma}^2$

$$\mathbb{E}\{\hat{\sigma}^2\} = \mathbb{E}\left\{\frac{\sum_{i=1}^N (\hat{A}_i - \hat{\mu})^2}{N}\right\} = \frac{1}{N}\mathbb{E}\left\{\sum_{i=1}^N [(A_i - \mu)^2 + \delta_i^2]\right\} = \sigma^2 + \sigma_\delta^2, \quad (5)$$

where $\sigma^2$ is the true floating-point variance of the batch and $\sigma_\delta^2$ is the variance of the noise introduced by the linear mapping to fixed-point. Note that the error variance $\sigma_\delta^2$ is rather small and can be integrated to $\epsilon$ in the denominator of equation (3).

## 4 Theoretical analysis of SGD using the proposed method

The generic equation of weight update in the $k^{\text{th}}$ iteration of SGD is

$$w_{k+1} = w_k + \alpha_k g(w_k, \xi_k), \tag{6}$$

where $g(w_k, \xi_k)$ is the estimated gradients of random samples of the batch generated by the seed $\xi_k$, and $\alpha_k$ is the learning rate. We make the following common assumption in the sequel.

**Assumption 1 (Lipschitz-continuity).** The loss function $\mathcal{L}(w)$ is continuously differentiable and its gradients $\nabla \mathcal{L}(w)$ satisfies the following inequality where $L > 0$ is the Lipchitz constant

$$\mathcal{L}(w) \leqslant \mathcal{L}(\bar{w}) + \nabla \mathcal{L}(\bar{w})^\top (w - \bar{w}) + \frac{1}{2} L ||w - \bar{w}||_2^2; \quad \forall\, w, \bar{w} \in \mathbb{R}^d. \tag{7}$$

**Assumption 2.** (i) $\mathcal{L}(w_k)$ is bounded. (ii) Estimated gradients $g(w_k, \xi_k)$ is an unbiased estimator of the true gradients of the loss function

$$\nabla \mathcal{L}(w_k)^\top \mathbb{E}_{\xi_k}\{g(w_k, \xi_k)\} = ||\nabla \mathcal{L}(w_k)||_2^2 = ||\mathbb{E}_{\xi_k}\{g(w_k, \xi_k)\}||_2^2,$$

and (iii,a) there exist scalars $M \geqslant 0$ and $M_V \geqslant 0$ such that for all iterations of SGD $\mathbb{V}_{\xi_k}\{g(w_k, \xi_k)\} \leqslant M + M_V ||\nabla \mathcal{L}(w_k)||_2^2$.

Note that here we define

$$\mathbb{V}_{\xi_k}\{g(w_k, \xi_k)\} := \mathbb{E}_{\xi_k}\{||g(w_k, \xi_k)||_2^2\} - ||\mathbb{E}_{\xi_k}\{g(w_k, \xi_k)\}||_2^2.$$

Also from *Assumption 2. (ii) and (iii,a)*, the second moment bound can be derived

$$\mathbb{E}_{\xi_k}\{||g(w_k, \xi_k)||_2^2\} \leqslant M + M_G ||\nabla \mathcal{L}(w_k)||_2^2 \quad \text{with } M_G := 1 + M_V. \tag{8}$$

**Effect of gradient variance on convergence:** The quality of the estimated gradients $g(w_k, \xi_k)$ directly affects the convergence of the SGD. The effect of first and second moments of gradient are already studied on *real numbers* in the literature.

**Lemma 1.** Suppose *Assumption 2* is true, then we have

$$\mathbb{E}_{\xi_k}\{\mathcal{L}(w_{k+1})\} - \mathcal{L}(w_k) \leqslant -(1 - \frac{1}{2}\alpha_k L M_G)\alpha_k ||\nabla \mathcal{L}(w_k)||_2^2 + \frac{1}{2}\alpha_k^2 L M. \tag{9}$$

*Proof:* See Bottou et al. [14, Lemma 4.4].

Inequality (9) shows the effect of gradient variance bounds, $M$ and $M_G$, on each iterate of SGD, and shows the greater the variance, the more deterioration in the quality of SGD steps. The first term, $-(1 - \frac{1}{2}\alpha_k L M_G)\alpha_k ||\nabla \mathcal{L}(w_k)||_2^2$ contributes to the decrease of the loss function while the second term, $\frac{1}{2}\alpha_k^2 L M$, prevents it. Upon choosing the correct gradient estimates, the right hand side of inequality (9) is bounded by a deterministic quantity, and asymptotically ensures sufficient descent of the loss $\mathcal{L}(w)$. Note that the expectation $\mathbb{E}_{\xi_k}$ is taken over random samples with seed $\xi_k$.

### 4.1 Fixed-point gradient noise

When performing representation mapping in the back-propagation, the quality of the gradients deteriorate. Thus, there is a need to consider the effect of fixed-point mapping variance. Then, *Assumption 2 (iii,a)* should be modified accordingly.

**Remark 1.** Note that *Assumption 2 (i), (ii)* still hold after fixed-point mapping because of stochastic rounding, i.e. the fixed-point gradient remains an unbiased estimator of gradient (refer to Appendix A.1).

**Assumption 2 (iii,b).** When the gradients are in fixed-point format i.e. $\hat{g}(w_k, \xi_k)$, there exist scalars $M \geqslant 0$, $M_V \geqslant 0$, $M^q \geqslant 0$ and $M_V^q \geqslant 0$ such that for all iterations of SGD

$$\mathbb{V}_{\xi_k}\{\hat{g}(w_k, \xi_k)\} \leqslant M + M^q + (M_V + M_V^q)||\nabla \mathcal{L}(w_k)||_2^2.$$

If $\tilde{M} := M + M^q$ and $\tilde{M}_V := M_V + M_V^q$ , then *Assumption 2 (iii,b)* takes the exact form of *Assumption 2 (iii,a)* i.e. $\mathbb{V}_{\xi_k}\{g(w_k, \xi_k)\} \leqslant \tilde{M} + \tilde{M}_V ||\nabla \mathcal{L}(w_k)||_2^2$. However, here we separated $M^q$ and $M_V^q$ to emphasize the effect of fixed-point mapping on the true gradients.

**Remark 2.** If *Assumption 2 (iii,b)* holds true, inequality (9) can be transformed to its fixed-point version

$$\mathbb{E}_{\xi_k}\{\mathcal{L}(w_{k+1})\} - \mathcal{L}(w_k) \leqslant -(1 - \frac{1}{2}\alpha_k L(M_G + M_G^q))\alpha_k ||\nabla \mathcal{L}(w_k)||_2^2 + \frac{1}{2}\alpha_k^2 L(M + M^q)$$
$$\text{with } M_G^q := 1 + M_V^q.$$
(10)

Inequality (10) shows the effect of **added** representation mapping variance with bounds, $M^q$ and $M_G^q$, on each iterate of SGD. This observation shows that fixed-point mapping degrades the convergence of SGD unless its variance bounds are relatively small, or controlled by the learning rate. As an example, refer to Appendix A.2 for analytical derivations of $M^q$ and $M_G^q$ for the back-propagation of a linear layer involving a fixed-point inner product.

## 4.2 Strongly convex and locally convex loss

**Assumption 3 (Strong convexity).** The loss function $\mathcal{L}(w)$ is differentiable and strongly convex. We recall that strong convexity for differentiable functions is equivalent to the following inequality with some constant $c > 0$

$$\mathcal{L}(w) \geqslant \mathcal{L}(\bar{w}) + \nabla \mathcal{L}(\bar{w})^\top (w - \bar{w}) + \frac{1}{2}c||w - \bar{w}||_2^2; \quad \forall w, \bar{w} \in \mathbb{R}^d. \tag{11}$$

A strongly convex function has a unique minimum point at $w_*$ with the loss value $\mathcal{L}_* = \mathcal{L}(w_*)$.

**Theorem 1.** Suppose *Assumptions 1, 2(i), 2(ii), 2 (iii,b), 3* are all true, then a SGD method running with fixed-point gradients i.e. $\hat{g}(w_k, \xi_k)$ and a fixed learning rate $0 < \bar{\alpha} \leqslant \frac{1}{L(M_G + M_G^q)}$ satisfies the following bound for its optimality gap with the minimum loss $\mathcal{L}_*$ at the $k^{\text{th}}$ iteration

$$\mathbb{E}\{\mathcal{L}(w_k) - \mathcal{L}_*\} \leqslant \frac{\bar{\alpha}L(M + M^q)}{2c} + (1 - \bar{\alpha}c)^{(k-1)}\left(\mathcal{L}(w_1) - \mathcal{L}_* - \frac{\bar{\alpha}L(M + M^q)}{2c}\right)$$
$$\xrightarrow{k \to \infty} \frac{\bar{\alpha}L(M + M^q)}{2c}.$$
(12)

*Proof.* See Appendix A.3.

**Remark 3.** Note that when $k \to \infty$, $\frac{\bar{\alpha}LM}{2c}$ is the original optimality gap (i.e. $\mathbb{E}\{\mathcal{L}(w_k) - \mathcal{L}_*\}$) with *real* gradients, see Bottou et al. [14, Theorem 4.6], and then, the optimality gap is also increased by $\frac{\bar{\alpha}LM^q}{2c}$ due to fixed-point representation. Here we argue that by keeping the variance bound $M^q$ relatively small via choosing the correct number format, we can theoretically achieve the original performance. On the other hand, optimality gap is related to $\bar{\alpha}$, which means smaller learning rates leads to smaller optimality gap.

**Remark 4 (Local convexity).** Strongly convex loss is not a realistic assumption in large deep learning models. However, local convexity is a more realistic assumption i.e. $\mathcal{L}$ is convex around the minimum point $w_*$. Hence, inequality (12) still holds around the minimum point $w_*$ of a locally convex loss. We essentially train our ImageNet ResNet18 classification in the same way: first we train the network with a fixed learning rate until reaching a certain optimality gap; then the learning rate is reduced to a smaller fixed value in order to shrink the optimality gap.

**Empirical evidence:** Figure 3(a) demonstrates the locally convex loss landscape of ResNet18 training on CIFAR10 dataset using floating-point computation. Figure 3(b) shows the same loss landscape in the fixed-point `int8` format. We perturbed the weights around the $w_*$ in $x$ and $y$ axes using Gaussian noise and evaluated the loss $\mathcal{L}$ in the $z$ axis to acquire Figures 3(a) and 3(b). Comparing these two figures pronounces our assumption of local convexity in both integer and floating-point tests. Figure 3(c) shows a comparison of the loss trajectory for floating-point and integer training. The fixed-point gradients are unbiased estimators of the true gradients, so the trajectory of the integer training closely follows the trajectory of its floating-point counterpart.

**Remark 5.** We implemented an integer weight update, hence, the computations of equation (6) is also performed in integer arithmetic. It is shown in Appendix A.4 that integer weight update with stochastic rounding is an unbiased estimator of the true weight update.

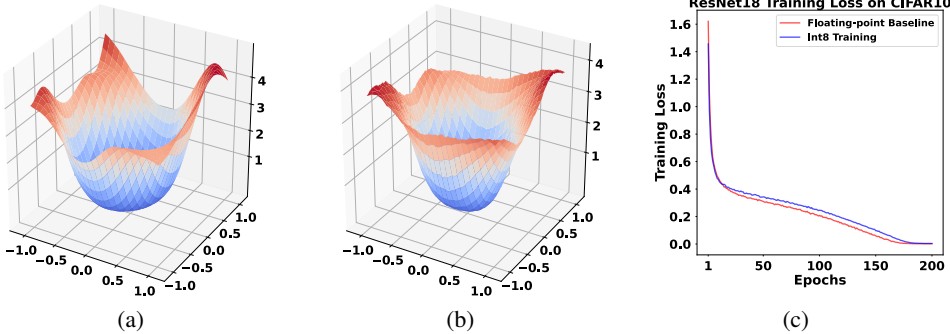

Figure 3: (a) Floating-point loss landscape, (b) Fixed-point `int8` loss landscape c) Training loss trajectory comparison. **Integer training setup:** `int8` linear layer, `int8` convolutional layer and `int8` batch-norm layer.

## 5 Experimental results

**Image classification:** Table 1 reports the experimental results of our proposed integer training method on the conventional vision classification models. In this set of models, we used `int8` linear layer, `int8` convolutional layer, `int8` batch-norm layer, and `int16` SGD to form a *fully integer training* pipeline. Note that all forward and back-propagation computations of layers are performed in *integer* format. Moreover, we have chosen smaller vision models such as ResNet18 and MobileNetV2 knowing that smaller models are harder to train. Our experimental results show that our proposed integer training method is as good as floating-point with negligible loss of accuracy ($\leqslant 0.5\%$ on ImageNet).

**Remark 6.** We compare our integer training results with official Pytorch floating-point training. For instance, in conventional vision models (Table 1), we compare the results directly with the accuracy of *torchvision* models that are reported on the official Pytorch website [1]. We further emphasize that there is no change in hyper-parameters of the training. The full set of hyper-parameters of our experiments are reported in Appendix A.5.

**Vision transformer:** We validated the applicability of our proposed integer training on the original vision transformer model, notably ViT-B-16-224 [15]. We took the floating point checkpoint pre-trained on ImageNet21K[2]. We used Huggingface [16] to fine-tune the model on CIFAR10. In this experiment, we used `int8` linear layer, `int8` matrix multiplication, `int8` convolutional layer, and `int8` layer-norm for our integer training pipeline. The result is reported in Table 1 demonstrating negligible loss of accuracy ($\leqslant 0.5\%$). Note that for this experiment, the computation of softmax in attention mechanism is in floating point.

**Semantic segmentation:** To validate our proposed integer training on semantic segmentation task, we used DeepLabV1/V2[3] with ResNet-101 as the backbone. We trained the models on PASCAL VOC-2012 [17] and MS COCO 10K [18]. For the PASCAL dataset, as suggested by Chen et al. [19], the models were initialized with MS COCO checkpoint and data augmentation was used during training. Likewise, for COCO dataset we used PASCAL checkpoint for initialization. The batch-norm layers are frozen in our experiments as suggested in [19]. We used linear and convolutional layers in `int8` format. Table 2 shows that the mean intersect of union (mIOU) of our method closely matches the floating-point baseline.

**Object detection:** We used Faster R-CNN[20] with ResNet50 as its backbone and Single Shot Detector (SSD)[21] which relies on VGG-16 for its backbone. In these experiments, we used `int8` linear and convolutional layers, while batch-norm layers are frozen. We integrated our integer training method with MMDetection[22] toolbox and performed our experiments on MS COCO 10K[18], PASCAL VOC-2007[23], VOC-2012[17], and Cityscapes[24] datasets. As shown in Table 3, the mean average precision (mAP) of our proposed integer training method is very close to floating-point baseline.

---

[1]Refer to Pytorch Models: https://pytorch.org/vision/stable/models.html
[2]Refer to: https://huggingface.co/google/vit-base-patch16-224
[3]Refer to : https://github.com/kazuto1011/deeplab-pytorch

Table 1: Classification

| Model | Dataset | Accuracy | |
| | | int8
(top1 - top5) | Pytorch baseline float
(top1 - top5) |
|---|---|---|---|
| Conventional Vision Models | | | |
| **ResNet18** | CIFAR10 | 94.84 - N/A | 95.34 - N/A |
| **ResNet18** | CIFAR100 | 75.38 - N/A | 76.14 - N/A |
| **ResNet18** | ImageNet | 69.25 - 88.79 | 69.75 - 89.07 |
| **MobileNetV2** | ImageNet | 72.80 - 90.83 | 71.87 - 90.28 |
| Vision Transformer | | | |
| **ViT-B** (fine-tuning) | CIFAR10 | 98.80 - N/A | 99.10 - N/A |

Table 2: Semantic Segmentation

| Method | Dataset | mIOU | |
| | | int8 | baseline
float |
|---|---|---|---|
| **DeepLab-V1** | VOC | 74.73 | 75.00 |
| | COCO | 34.80 | 34.70 |
| **DeepLab-V2** | VOC | 77.65 | 77.71 |
| | COCO | 35.90 | 36.00 |

Table 3: Object Detection

| Method | Dataset | mAP | |
| | | int8 | baseline
float |
|---|---|---|---|
| **Faster R-CNN** | COCO | 37.40 | 37.80 |
| | VOC07+12 | 80.14 | 80.31 |
| | Cityscapes | 39.50 | 40.00 |
| **SSD** | COCO | 42.50 | 43.60 |

Table 4: Comparison with State of the Art

| Model | Dataset | Method | | | | |
| | | Ours | [2] | [3] | [4] | [6] |
|---|---|---|---|---|---|---|
| **MobileNetV2** | **ImageNet** | 72.8 | 70.5 | 71.9 | 71.2 | 72.6 |
| **ResNet18** | **ImageNet** | 69.3 | - | 70.2 | 69.7 | 71.1 |
| **DeepLab-V1** | **VOC** | 74.7 | 69.9 | - | - | - |
| **Faster R-CNN** | **COCO** | 37.4 | - | 37.4 | 34.9 | - |

Table 5: Low-bit Integer Training

| Model | Dataset | bit-width | | | | |
| | | int8 | int7 | int6 | int5 | int4 |
|---|---|---|---|---|---|---|
| **ResNet18** | **CIFAR10** | 94.8 | 94.7 | 94.47 | 88.5 | Diverges |

**Comparison with state of the art:** Table 4 provides a comparison between our training method and state of the art across different experiments. There are some important differences between our proposed method and other works: (i) our proposed integer training method uses a fixed-point batch-norm layer where both forward and back propagation is computed using integer arithmetic, (ii) our proposed training method uses an integer-only SGD, (iii) in our training method, no hyper-parameter is changed while other methods have changed hyper-parameters or used gradient clipping.

**Low-bit integer training:** Table 5 provides an ablation study of how lowering integer bit-width can affect the training accuracy. Our experiments shows that training has a significant drop of accuracy with `int5` and diverges using `int4` number formats.

# 6  Conclusion

We proposed a hardware-friendly integer training method based on coupling a linear fixed-point mapping with a non-linear inverse mapping. This method performs the representation mapping directly on the floating-point number format. Furthermore, our method uses stochastic rounding and produces unbiased estimators of gradients in the back-propagation. Thus, benefiting from unbiased

gradients and effective representation mapping, the training loss trajectory closely follows its floating-point version. Moreover, there is no need to tune hyper-parameters or perform gradient clipping to correct the inappropriate gradient and weight updates. Using our proposed technique, we designed the integer version of the most vital components of deep learning such as linear layer, convolutional layer, batch-norm, layer-norm, and stochastic gradient descent (SGD). Our experimental results show the effectiveness of the proposed method, where the accuracy loss is negligible in a wide variety of the training tasks. Furthermore, in our classification tests all training components are in integer arithmetic. We theoretically studied the effect of our proposed method on SGD, and demonstrated why the optimality gap of convergence is shifted by a negligible amount.

## Acknowledgments and Disclosure of Funding

The authors would like to thank Richard Wu and Vanessa Courville for their constructive comments. The authors would like to thank Sara Elkerdawy for initiating the object detection and semantic segmentation experiments during her internship at Huawei.

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
