# A Appendix

## A.1 Stochastic Rounding

In this paper, stochastic rounding for a variable $x$ where $x_1 < x < x_2$ is defined as

$$\hat{x} = \begin{cases} x_1 & \text{with probability } \frac{x_2 - x}{x_2 - x_1} \\ x_2 & \text{with probability } \frac{x - x_1}{x_2 - x_1} \end{cases} \tag{13}$$

it is easy to see that $\hat{x}$ is an unbiased estimator of $x$:

$$\mathbb{E}\{\hat{x}\} = x_1 \frac{x_2 - x}{x_2 - x_1} + x_2 \frac{x - x_1}{x_2 - x_1} = x \tag{14}$$

and if we relate $x$ and $\hat{x}$ using an error term $\delta$ (i.e. $\hat{x} = x + \delta$), then $\mathbb{E}\{\delta\} = 0$.

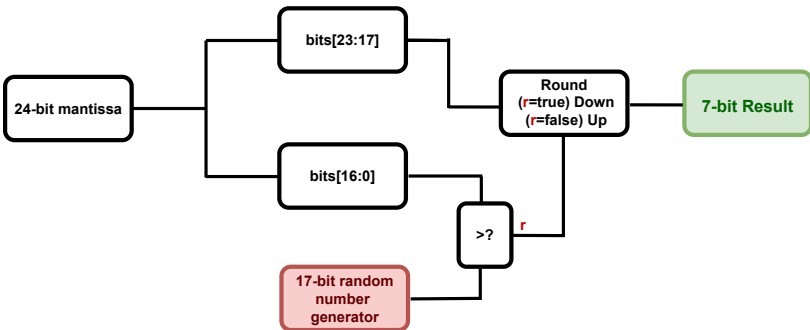

Figure 4: An implementation of stochastic rounding.

A realization of the stochastic rounding is shown in Figure 4. Here, a 24-bit single floating-point mantissa (including implicit hidden bit) is rounded stochastically to a 7-bit value. In this figure, the direction of rounding is determined by comparing a random number that is generated on-the-fly with the lower 17-bit of the mantissa.

## A.2 Representation mapping increases the gradients variance: Linear layer example

A linear layer is essentially a matrix multiplication. Let us denote $X$ as inputs, $W$ as weights and $Y$ as the output of a linear layer where $Y = XW$. Following the notation of this paper, we also denote our fixed-point version of this layer as $\hat{Y} = \hat{X}\hat{W}$.

In our proposed integer back-propagation, the layer receives the upstream gradient $\hat{G} := \frac{\partial \hat{L}}{\partial \hat{Y}}$. Using the chain-rule, the gradient with respect to weights is

$$\frac{\partial \hat{L}}{\partial \hat{W}} = \frac{\partial \hat{Y}}{\partial \hat{W}} \frac{\partial \hat{L}}{\partial \hat{Y}} = \hat{X}^{\top} \frac{\partial \hat{L}}{\partial \hat{Y}} = \hat{X}^{\top} \hat{G}. \tag{15}$$

$$\mathbb{V}\{\hat{C}_{ij}\} = \mathbb{V}\left\{\sum_{k=1}^{K}\hat{X}_{ik}^{\top}\hat{G}_{kj}\right\} = \sum_{k=1}^{K}\mathbb{V}\left\{\hat{X}_{ik}^{\top}\hat{G}_{kj}\right\} + \sum_{k=1}^{K}\sum_{\substack{k'=1\\k\neq k}}^{K}\mathbb{COV}\left(\hat{X}_{ik}^{\top}\hat{G}_{kj}, X_{ik'}^{\top}\hat{G}_{k'j}\right)$$

$$= \sum_{k=1}^{K}\left\{\mathbb{E}\{(\hat{X}_{ik}^{\top})^2\}\mathbb{E}\{(\hat{G}_{kj})^2\} - \mathbb{E}^2\{(\hat{X}_{ik}^{\top})\}\mathbb{E}^2\{(\hat{G}_{kj})\}\right\} + \sum_{k=1}^{K}\sum_{\substack{k'=1\\k\neq k}}^{K}\mathbb{COV}\left(\hat{X}_{ik}^{\top}\hat{G}_{kj}, X_{ik'}^{\top}\hat{G}_{k'j}\right)$$

$$= \sum_{k=1}^{K}\left\{\mathbb{E}\{(X_{ik}^{\top}+\delta_{ik}^{X})^2\}\mathbb{E}\{(G_{kj}+\delta_{kj}^{G})^2\} - \mathbb{E}^2\{(X_{ik}^{\top}+\delta_{ik}^{X})\}\mathbb{E}^2\{(G_{kj}+\delta_{kj}^{G})\}\right\}$$

$$+ \sum_{k=1}^{K}\sum_{\substack{k'=1\\k\neq k}}^{K}\mathbb{COV}\left(X_{ik}^{\top}G_{kj}, X_{ik'}^{\top}G_{k'j}\right)$$

$$\leqslant \sum_{k=1}^{K}\left\{\mathbb{V}\{X_{ik}^{\top}G_{kj}\} + \sigma_X^2\mathbb{E}\{G_{kj}^2\} + \sigma_G^2\mathbb{E}\{X_{ik}^{\top 2}\} + \sigma_X^2\sigma_G^2\right\} + \sum_{k=1}^{K}\sum_{\substack{k'=1\\k\neq k}}^{K}\mathbb{COV}\left(X_{ik}^{\top}G_{kj}, X_{ik'}^{\top}G_{k'j}\right)$$

$$= \mathbb{V}\left\{\sum_{k=1}^{K}X_{ik}^{\top}G_{kj}\right\} + \sigma_G^2\mathbb{E}\{||X_{i,}^{\top}||_2^2\} + \sigma_X^2\mathbb{E}\{||G_{,j}||_2^2\} + K\sigma_X^2\sigma_G^2$$

$$= \mathbb{V}\{C_{ij}\} + \sigma_G^2\mathbb{E}\{||X_{i,}^{\top}||_2^2\} + \sigma_X^2\mathbb{E}\{||G_{,j}||_2^2\} + K\sigma_X^2\sigma_G^2. \tag{16}$$

Note that in inequality (16), $\sigma_G^2 = \max(\mathbb{V}\{\delta_{i,j}^{G}\})$ knowing that error terms $\delta_{i,j}^{G}$ have essentially similar distributions. Likewise, $\sigma_X^2 = \max(\mathbb{V}\{\delta_{i,j}^{X}\})$. Also note that $X$ and $G$ are matrices of random variables in the back-propagation and $||X_{i,}^{\top}||_2^2$ denotes the norm-2 of the $i^{\text{th}}$ *row* of $X^{\top}$ and $||G_{,j}||_2^2$ denotes the norm-2 of the $j^{\text{th}}$ *column* of $G$. Also, $\hat{X}_{ik}^{\top}$ denotes $ik^{\text{th}}$ element of the matrix $\hat{X}^{\top}$.

By having the following definitions

$$\begin{cases} M^q := \sigma_G^2\mathbb{E}\{||X_{i,}^{\top}||_2^2\} + K\sigma_X^2\sigma_G^2 \\ M_V^q := \sigma_X^2 \end{cases} \tag{17}$$

we can re-organize the inequality (16) as

$$\mathbb{V}\{\hat{C}_{ij}\} \leqslant \mathbb{V}\{C_{ij}\} + M_V^q\mathbb{E}\{||G_{,j}||_2^2\} + M^q$$

$$\xrightarrow{\text{Assumption 2.(iii,a)}} \mathbb{V}\{\hat{C}_{ij}\} \leqslant (M_V + M_V^q)\mathbb{E}\{||G_{,j}||_2^2\} + (M + M^q). \tag{18}$$

**Remark.** Inequality (18) supports our *Assumption 2 (iii,b)* i.e.

$$\mathbb{V}_{\xi_k}\{\hat{g}(w_k, \xi_k)\} \leqslant M + M^q + (M_V + M_V^q)||\nabla\mathcal{L}(w_k)||_2^2$$

for considering the effect of our representation mapping method on gradients variance.

## A.3  Proof of *Theorem 1*

The proof goes along the proof of Bottou et al. [14, Theorem 4.6] in the case that the gradient variance bound increased as stated in *Assumption 2 (iii, b)*.

A convex function satisfying the inequality (11) given $\bar{w}, w \in \mathbb{R}^d$ represents a quadratic model

$$q(\bar{w}) := \mathcal{L}(w) + \nabla\mathcal{L}(w)^{\top}(\bar{w} - w) + \frac{1}{2}c||\bar{w} - w||_2^2, \tag{19}$$

and has a unique minimizer at $w_*$

$$w_* := w - \frac{1}{c}\nabla\mathcal{L}(w)$$

$$q(w_*) = \mathcal{L}(w) - \frac{1}{2c}||\nabla\mathcal{L}(w)||_2^2 \tag{20}$$

$$\to 2c(\mathcal{L}(w) - \mathcal{L}_*) \leqslant ||\nabla\mathcal{L}(w)||_2^2; \quad \forall w \in \mathbb{R}^d.$$

Also remember that the fixed learning rate in our integer back-propagation has the following constraint

$$0 < \bar{\alpha} \leqslant \frac{1}{L(M_G + M_G^q)}. \tag{21}$$

Starting from inequality (10) and using inequalities (20) and (21) we can write

$$\mathbb{E}_{\xi_k}\{\mathcal{L}(w_{k+1})\} - \mathcal{L}(w_k) \leqslant - (1 - \frac{1}{2}\bar{\alpha}L(M_G + M_G^q))\bar{\alpha}||\nabla\mathcal{L}(w_k)||_2^2 + \frac{1}{2}\bar{\alpha}^2 L(M + M^q)$$

$$\leqslant - \frac{1}{2}\bar{\alpha}||\nabla\mathcal{L}(w_k)||_2^2 + \frac{1}{2}\bar{\alpha}^2 L(M + M^q)$$

$$\leqslant - \bar{\alpha}c(\mathcal{L}(w_k) - \mathcal{L}_*) + \frac{1}{2}\bar{\alpha}^2 L(M + M^q). \tag{22}$$

By subtracting $\mathcal{L}_*$, rearrange, and taking total expectation from both sides of inequality (22) we have

$$\mathbb{E}\{\mathcal{L}(w_{k+1}) - \mathcal{L}_*\} \leqslant (1 - \bar{\alpha}c)\mathbb{E}\{\mathcal{L}(w_k) - \mathcal{L}_*\} + \frac{1}{2}\bar{\alpha}^2 L(M + M^q). \tag{23}$$

Then by subtracting $\frac{\bar{\alpha}L(M+M^q)}{2c}$ from both sides

$$\mathbb{E}\{\mathcal{L}(w_{k+1}) - \mathcal{L}_*\} - \frac{\bar{\alpha}L(M + M^q)}{2c} \leqslant (1 - \bar{\alpha}c)\mathbb{E}\{\mathcal{L}(w_k) - \mathcal{L}_*\} + \frac{1}{2}\bar{\alpha}^2 L(M + M^q) - \frac{\bar{\alpha}L(M + M^q)}{2c}$$

$$= (1 - \bar{\alpha}c)\left(\mathbb{E}\{\mathcal{L}(w_k) - \mathcal{L}_*\} - \frac{\bar{\alpha}L(M + M^q)}{2c}\right). \tag{24}$$

Thus, *Theorem 1* can be proven by applying inequality (24) repeatedly for $k \in \mathbb{N}$. Also note that using inequality (21) it is easy to derive that $0 < (1 - \bar{\alpha}c) < 1$ because

$$0 < \bar{\alpha}c \leqslant \frac{c}{L(M_G + M_G^q)} \leqslant \frac{c}{L} \leqslant 1, \tag{25}$$

hence, in *Theorem 1*, if $k \to \infty$, then $(1 - \bar{\alpha}c)^k \to 0$ and the optimality gap of integer training algorithm using inequality (12) is

$$\mathbb{E}\{\mathcal{L}(w_k) - \mathcal{L}_*\} \leqslant \frac{\bar{\alpha}L(M + M^q)}{2c} \tag{26}$$
$$\text{s.t. } k \to \infty.$$

### A.4 Integer weight update

In the integer weight update, the equation (6) transforms to its fixed-point version with integer-only arithmetic

$$\hat{w}_{k+1} = \hat{w}_k + \hat{\alpha}_k\hat{g}(w_k, \xi_k). \tag{27}$$

By expanding the error terms for the representation mapping and taking expatiation on both sides we can see the weights are on the average updated equivalent to the true wights.

$$\mathbb{E}\{\hat{w}_{k+1}\} = \mathbb{E}\{\hat{w}_k + \hat{\alpha}_k\hat{g}(w_k, \xi_k)\}$$
$$= \mathbb{E}\{w_k + \delta_k^w + (\alpha_k + \delta^\alpha)(g(w_k, \xi_k) + \delta^g)\}$$
$$= w_k + \alpha_k g(w_k, \xi_k)$$
$$= w_{k+1} \tag{28}$$

We used stochastic rounding for our weight update operation, thus $\mathbb{E}\{\delta\} = 0$ and using our proposed method, $\hat{g}$ is unbiased estimator of $g$.

### A.5 Experimental setup

**Computing resources:** We ran our experiments using an in-house developed integer emulator to avoid common floating-point quantization techniques. In our emulator, we perform the *representation mapping* within the GPU memory. We further developed the integer deep learning modules using Pytorch autograd functionality on top of our integer emulator. Experimental results of this paper are run using the following number of GPUs.

- ResNet18 on ImageNet requires 4×V100 GPUs when batch size is 512 and 2×V100 GPUs when batch size is 256.
- MobileNetV2 on ImageNet requires 8×V100 GPUs when batch size is 512.
- ResNet18 on CIFAR10 runs on 1×V100 GPUs when batch size is 128.
- Semantic segmentation experiments require 2×V100 GPUs.
- Object detection experiments require 8×V100 GPUs.
- ViT-B fine-tuning experiment requires 8×V100 GPUs.

**Classification:** The hyper-parameters of our classification experiments are reported in Table 6. We used SGD with momentum of 0.9 for conventional classification experiments and AdamW for ViT fine-tuning.

Table 6: Hyper-parameters for classification experiments

| Dataset | Model | Training epochs | Learning rate | LR scheduling | Weight decay | Batch size |
|---------|-------|-----------------|---------------|---------------|--------------|------------|
| ImageNet | ResNet18 | 90 or 100 | 0.1 | ×0.1 every 30 epochs | 1e-4 | 512 |
| | MobileNetV2 | 70 | 0.1 | Cosine with $T_{max} = 70$ | 4e-5 | 512 |
| CIFAR10 | ResNet18 | 200 | 0.1 | Cosine with $T_{max} = 90$ | 5e-4 | 128 |
| | ViT-B fine-tuning | 100 | 5e-5 | Cosine with $T_{max} = 100$ | 0.01 | 512 |
| CIFAR100 | ResNet18 | 200 | 0.1 | Reduce at epochs 80 and 120 | 1e-4 | 128 |

**Semantic segmentation:** The hyper-parameters for the semantic segmentation experiments are provided in Table 7. For DeepLabV1 and DeepLabV2 one-scale and multi-scale loss was used respectively. Conditional random field post-processing was used at the network's output as proposed in Chen et al. [19]. In all the experiments we use SGD with momentum of 0.9 and weight decay of $5 \times 10^{-4}$.

Table 7: Hyper-parameters for semantic segmentation experiments

| Model | Dataset | Training iterations | Learning rate | Batch size | Data Augmentation | CRF |
|-------|---------|---------------------|---------------|------------|-------------------|-----|
| DeepLabV1/V2 | VOC | 20000 | 2.5e-4 | 16 | ✓ | ✓ |
| | COCO | 30000 | 2.5e-4 | 16 | - | ✓ |

**Object detection:** The hyper-parameters are provided in Table 8. All experiments use an SGD optimizer with a momentum of 0.9 and a weight decay parameter of $10^{-5}$. The experiments indicated with *LR Warmup* use a linear warm-up function with a ratio of $10^{-3}$ for the first 500 iterations. For the rest of training, the learning rate is reduced by 0.1 at the epochs indicated in the *LR Reduction Epochs* column.

Table 8: Hyper-parameters for object detection experiments

| Model | Dataset | Training epochs | Learning rate | Per GPU Batch size | LR reduction epochs | LR Warmup |
|-------|---------|-----------------|---------------|--------------------|---------------------|-----------|
| Faster R-CNN | COCO | 12 | 0.2 | 1 | 8 and 11 | ✓ |
| | VOC | 12 | 0.1 | 2 | 9 | - |
| | Cityscapes | 64 | 0.1 | 1 | 56 | ✓ |
| SSD | COCO | 24 | 0.2 | 1 | 16 and 22 | ✓ |

### A.6 Background on quantization methods

Symmetric uniform quantization is normally used in the literature of quantized back-propagation (see [2, 4, 3] ) to convert the gradients to 8-bit integers. Suppose scale $s = \max(|x|)$, then

$$x_{\text{quantized}} = \text{round}(127. \frac{\text{clamp}(x, s)}{s}),$$

with the clipping function $\text{clamp}(x, s)$ defined as:

$$\text{clamp}(x, s) = \begin{cases} x & |x| \leq s \\ \text{sign}(x)s & |x| > s \end{cases}.$$

Furthermore, given the scale factor $s$, the de-quantization is performed as:

$$\hat{x}_{\text{float}} = x_{\text{quantized}} \times \frac{s}{127}.$$

We further emphasize here that our method of representation mapping differs from this quantization method, where we directly manipulate the floating point number format as mentioned in Section 3.1.