# OpenReview forum: "Is Integer Arithmetic Enough for Deep Learning Training?"
_NeurIPS.cc/2022/Conference — NeurIPS 2022 Accept_

### Official Review · Reviewer_m2A4 · 2022-06-21

**Rating:** 5
**Confidence:** 4
**Soundness:** 3 good
**Presentation:** 2 fair
**Contribution:** 3 good

**Summary:**

The paper provides mathematical analysis and experimental results to demonstrate an INT8 neural network that are trained with int8 arithmetic. The proposed method is hardware friendly and does not rely on training data and weights distribution.

**Questions:**

N/A

**Limitations:**

Yes.

**Strengths And Weaknesses:**

Strengths
1. The proposed method is interesting and the mathematical deduction is sound to me.
2. The results of the compact models are comparable with the FP32 models.

Weaknesses
1. Please compare with other INT8 methods instead of only FP32 models.
2. The title and abstract are misleading to me. There are also floating-point arithmetic in the model, so it is not integer arithmetic only.
3. I suggest to replace cifar10 experiments with cifar100. Though, ImageNet experiments can demonstrate the method's effectiveness on large datasets.

---

> ### Author Response · Authors · 2022-08-02
> **Response to Reviewer m2A4**
>
> **Question/Concern: Please compare with other INT8 methods instead of only FP32 models.**
>
> Thanks for your comments, we added Table.4 and also its corresponding paragraph in **lines 297-302**
>
> |**Model**                 | **Dataset**              | **Ours** | **[1]** | **[2]** | **[3]** | **[4]**|
> |---                | ---              | --- | --- | --- | --- | ---
> **MobileNetV2**  | **ImageNet**    | 72.8 | 70.5    | 71.9    | 71.2    | 72.6
> **ResNet18**     | **ImageNet**    | 69.3 | -       | 70.2    | 69.7    | 71.1
> **DeepLab-V1**   | **VOC**         | 74.7 | 69.9    | -       | -       | -
> **Faster R-CNN** | **COCO**        | 37.4 | -       | 37.4    | 34.9    | -
>
> *Comparison with state of the art:  Table 4 provides a comparison between our training method and state of the art across different experiments. There are some important differences between our proposed method and other works: (i) our proposed integer training method uses a fixed-point batch-norm layer where both forward and back propagation is computed using integer arithmetic, (ii) our proposed training method uses an integer-only SGD,  (iii) in our training method, no hyper-parameter is changed while they have changed hyper-parameters or used gradient clipping.*
>
>
> **Question/Concern: The title and abstract are misleading to me. There are also floating-point arithmetic in the model, so it is not integer arithmetic only.**
>
> The training of Conventional Vision Models in Table 1 is performed using integer arithmetic pipeline. That is why, we strongly believe that integer arithmetic for training deep learning models suffices.
> We acknowledge the fact that aside from ReLU, integer arithmetic for other non-linear functions such as attention mechanism has not been considered in our paper. Having said this, integer arithmetic for attention mechanism has recently been explored in other works [5]. The integer arithmetic of their gradients remains however to be thoroughly and systematically studied. We have added clarification on line 285 for attention mechanism in our revised manuscript.
>
> Following your comment we have toned down the abstract. Also please note that the title includes the question mark to hedge this claim.
>
> **Question/Concern: I suggest to replace cifar10 experiments with cifar100. Though, ImageNet experiments can demonstrate the method's effectiveness on large datasets.**
>
> As you mentioned, our ImagNet experiments can demonstrate capabilities of our proposed method.  Having said this, we are currently running the CIFAR100 tests with our emulator and we will add them to the paper as soon as possible.
>
> **References:**
>
> [1] Zhang, Xishan, et al. "Fixed-point back-propagation training." Proceedings of the IEEE/CVF Conference on Computer Vision and Pattern Recognition. 2020.
>
> [2] Zhao, Kang, et al. "Distribution adaptive int8 quantization for training cnns." Proceedings of the AAAI Conference on Artificial Intelligence. Vol. 35. No. 4. 2021.
>
> [3] Zhu, Feng, et al. "Towards unified int8 training for convolutional neural network." Proceedings of the IEEE/CVF Conference on Computer Vision and Pattern Recognition. 2020.
>
> [4] Jin, Qing, et al. "F8Net: Fixed-Point 8-bit Only Multiplication for Network Quantization." arXiv preprint arXiv:2202.05239 (2022).
>
> [5] Sari, Eyyüb, Vanessa Courville, and Vahid Partovi Nia. "IRNN: Integer-only recurrent neural network." ICPRAM 11 (2022)

---

> > ### Author Response · Authors · 2022-08-09
> > **New results for CIFAR100**
> >
> > Thank you for your comment, we have added the results of **CIFAR100** experiments to **Table 1** for your reference. We would appreciate if you increase your score if you are satisfied with our responses.
> >
> > | **Model**   | **Dataset**  |  int8  \\ (top1 - top5) |      Pytorch baseline float \\ (top1 - top5)|
> > | ---  | ---  |  :----: |  :----: |
> > | ***Conventional Vision Models***  |   |   |   |
> > ResNet18 | CIFAR10  |  94.84 - N/A   & 95.34 - N/A |
> > **ResNet18**  | **CIFAR100** | **75.38 - N/A**   | **76.14 - N/A** |
> > ResNet18     | ImageNet       | 69.25 - 88.79 | 69.75 - 89.07 |
> > MobileNetV2     | ImageNet       |  72.80 - 90.83  | 71.87 - 90.28 |
> > | ***Vision Transformer***     |
> > ViT-B fine-tuning    | CIFAR10 |   98.80 - N/A|     99.10 - N/A|

---

### Official Review · Reviewer_jC5f · 2022-07-05

**Rating:** 8
**Confidence:** 4
**Soundness:** 3 good
**Presentation:** 3 good
**Contribution:** 3 good

**Summary:**

* The authors have proposed a new integer arithmetic for neural network training. What's different from the previous work is that instead of using the scale from the linear floating point to integer mapping they used the maximum floating-point exponent as the scale. The mapping between the integer and floating point values is thus dramatically different from the existing quantization approaches.
* Previous approaches tried to keep both the forward propagation and the back propagation entirely in the int8 domain, but the training suffered from significant accuracy loss. With the new integer arithmetic proposed in this paper which also keeps the forward propagation and the back propagation entirely in the int8 domain, a few experiments have been performed on vision models demonstrating that the new approach sacrifices negligible accuracy.
* Theoretical analysis has also been performed on the new approach.


**Questions:**

* Line 33 why the dynamic range for int8 is [-127, 126] instead of [-128, 127]? Please put some explanations somewhere in the paper.
* When describing mantissa, please align with the IEEE standards. For example, for FP32 values, the bit-width of mantissa is 23 instead of 24. I understand that the authors are counting the leading hidden bit. But please describe it explicitly in the paper.
* There is a lack of clearance in explaining the stochastic rounding of mantissas, i.e., how the the 24-bit single floating-point mantissas are further rounded to 7-bit mantissas. If it is possible, please provide a few concrete examples in the paper, just like the one in line 147. The stochastic rounding is a key for the proofs shown later in the paper. If $\mathbb{E}\\{\hat{A}_i\\} = A_i$ cannot be well justified by the stochastic rounding, most of the proofs in the paper will be invalid.
* If shifting bits in Figure 1(a) is linear mapping, why the alignment module in Figure 1(b) which also shift bits is non-linear? The naming was a little bit confusing.
* Please also show a table of comparing the proposed approach with the previous "SOTA"s in the paper. I believe this is beneficial for the paper because previous work does not always stay in the int8 domain for training.
* Please also discuss about how even lower bit-width, such as 4-bit, will affect the training accuracy. Depending on the authors' implementation, it is possible to test it in the experiments as well.
* In section 4.1, assumption 2 (iii, b), is it possible that $M \geq 0$, $M^q \leq 0$ and $M + M^q \geq 0$? Same for $M_V$ and $M_V^q$. If it is possible, it might alter the remark 2 and 3. If it is not possible, please elaborate in the paper.
* Line 235, with "some" constant $c > 0$.
* The authors claimed that "by keeping the variance bound $M^q$ relatively small, we can theoretically achieve the original performance". How could we measure or ensure this for training different neural networks in practice? In line 53, the authors claimed that the optimality gap is only shifted by a negligible amount. I did not see the concrete amount and how the amount is measured.



**Limitations:**

* Although this work is targeted for neural network training, the inference arithmetic is slightly different from the existing popular symmetric quantization arithmetic, which makes most of the existing integer inference frameworks incompatible with the new approach. The author should discuss about the impact of the new approach to the existing inference infrastructures, since it will require new software to make it efficient.

**Strengths And Weaknesses:**

Strengths
* The approach proposed is simple yet seems surprisingly effective in practice.
* The authors were aware of the limitations of the assumptions for the proofs. The proofs are relatively speaking clean.

Weaknesses
* This is not necessarily a weakness. As I mentioned in the following discussions, there are a few places in the paper that can be further elaborated and clarified to make the paper even better.

---

> ### Author Response · Authors · 2022-08-02
> **Response to Reviewer jC5f**
>
> **Q: Line 33 why the dynamic range for int8 is ...**:
>
> Thank you for your thorough reading of our article, this is a typo, we corrected it in the revised version of the manuscript.
>
> **Q: When describing mantissa, please align with the IEEE standards...**
>
>  Thanks, the confusion for mantissa size is clarified in lines 142-143 of the revised manuscript.
>
> **Q: There is a lack of clearance in explaining the stochastic rounding**
>
>  As you mentioned, using stochastic rounding is one of the foundations of this paper's proof. We want to emphasize that in Appendix A.1 we provided the mathematical proof of why $\mathbb{E}{(\hat{A}_i)} = A_i$.
> Stochastic rounding is to round to either of the two nearest 1-digit numbers with
> a probability that depends on the distances to those numbers. As an example we define $A_i= 1 +
> 0.1$ as 1 with probability 0.9 and as 2 with probability 0.1, then the expected result is $\mathbb{E}(\hat A_i)= 0.9 \times 1 +
> 0.1 \times 2 = 1.1$, which is the exact answer.
> Also please note that our paper is not the first paper that uses stochastic rounding, for example references [2] and [3] cited on page 2 of our manuscript also used stochastic rounding.
>
> **Q: If shifting bits in Figure 1(a) is linear mapping, why the alignment module in Figure 1(b) which also shift bits is non-linear?**
>
>  This is indeed a very interesting question. In the linear mapping, all the elements of the integer tensor are shifted/rounded to 8-bit integer. Note that at this stage, all the shifts are ***right shift***. In the second stage or non-linear inverse mapping, we have integer values that are undergone some computations, some of them might become zero and some might overflow. The non-linear inverse mapping module take care of each element of tensor and shift it left or right according the status of that element. In this case, some elements of tensor might be shifted to right, some might be shifted to left and some might remain untouched. This is why we call it non-linear inverse mapping since the shift is not in a uniform direction for the whole tensor.
>
> **Q: Please also show a table of comparing the proposed approach with the previous "SOTA" ...**
>
> Thanks for your comments, we added Table.4 and also its corresponding paragraph in lines 297-302:
>
> |**Model**                 | **Dataset**              | **Ours** | **[1]** | **[2]** | **[3]** | **[4]**|
> |---                | ---              | --- | --- | --- | --- | ---
> **MobileNetV2**  | **ImageNet**    | 72.8 | 70.5    | 71.9    | 71.2    | 72.6
> **ResNet18**     | **ImageNet**    | 69.3 | -       | 70.2    | 69.7    | 71.1
> **DeepLab-V1**   | **VOC**         | 74.7 | 69.9    | -       | -       | -
> **Faster R-CNN** | **COCO**        | 37.4 | -       | 37.4    | 34.9    | -
>
> *Comparison with state of the art:  Table 4 provides a comparison between our training method and state of the art across different experiments. There are some important differences between our proposed method and other works: (i) our proposed integer training method uses a fixed-point batch-norm layer where both forward and back propagation is computed using integer arithmetic, (ii) our proposed training method uses an integer-only SGD,  (iii) in our training method, no hyper-parameter is changed while they have changed hyper-parameters or used gradient clipping.*
>
>
> **Q: Please also discuss about how even lower bit-width ...**
>
>  We have performed some experiment on 4 and 5-bits integer numbers, 5-bit integer  has around 7\%  accuracy drop on CIFAR 10 while 4-bit sometimes diverges.
>
> **Q: In section 4.1, assumption 2 (iii, b) ... **
>
>  $M^q$ and $M^q_v$ are quantities that are related to variance, so they cannot be negative. This point is clarified in *Assumption (iii,b)*.
>
> **Q: Line 235, with "some" constant $c > 0$**
>
> Thanks for thorough reading our paper, this typo is corrected in the revised version.
>
> **Q: The authors claimed that "by keeping the variance bound $M^q$  relatively small **
>
>  The loss values of integer and floating-point training are close only if the effect of integer representation mapping is negligible. This is reflected in $M^q$ which measures variations of fixed point gradient noise. As shown in Figure 3.c of our manuscript, The trajectory of loss for integer and floating-point training are very close. This confirms that $M^q$ is negligible for this experiment.
>
> **Q: The author should discuss about the impact of the new approach to the existing inference infrastructures**
>
> We do agree that in order to have an integer training framework, an efficient software/hardware co-design is necessary. NVIDIA A100 GPU which support int8 tensor cores is very much suited for these types of algorithms. However, we did not have access to them to evaluate our method. In this paper, our focus is to propose an efficient methodology as well as theoretical support for integer training. The implementation aspect of this method is left for future work.

---

> > ### Comment · Reviewer_jC5f · 2022-08-04
> > **A Good Paper But Still Needs Improvement**
> >
> > > As you mentioned, using stochastic rounding is one of the foundations of this paper's proof. We want to emphasize that in Appendix A.1 we provided the mathematical proof of why
> > . Stochastic rounding is to round to either of the two nearest 1-digit numbers with a probability that depends on the distances to those numbers. As an example we define  as 1 with probability 0.9 and as 2 with probability 0.1, then the expected result is
> > , which is the exact answer. Also please note that our paper is not the first paper that uses stochastic rounding, for example references [2] and [3] cited on page 2 of our manuscript also used stochastic rounding.
> >
> > Thanks for providing more clarifications. I did read A.1 previously but I had to guess how it would align into the context of the mantissas rounding. That was also why I explicitly requested to have a simple example in the main text without asking the reader to further check the literature. So let me just put how I thought it works in the context.
> >
> > Say after mantissas bit shifting, the 24-bit mantissas is
> >
> > $$
> > x = \underbrace{001011101010101010100000}_{\text{24}}
> > $$
> >
> > what's the value for $x_1$ and $x_2$? Is it
> >
> > $$
> > x_1 = \underbrace{00101110}_{\text{8}}
> > $$
> >
> > $$
> > x_2 = \underbrace{00101111}_{\text{8}}
> > $$
> >
> > $$
> > x_2 - x_1 = (0.0101110)_2 - (0.01011101010101010100000)_2
> > $$
> >
> > ?
> >
> > The authors should have included all of these in the main text. Otherwise there can be multiple different interpretations of what's going on there. A good paper explain key things in itself and the reader wouldn't have to guess.
> >
> > > This is indeed a very interesting question. In the linear mapping, all the elements of the integer tensor are shifted/rounded to 8-bit integer. Note that at this stage, all the shifts are right shift. In the second stage or non-linear inverse mapping, we have integer values that are undergone some computations, some of them might become zero and some might overflow. The non-linear inverse mapping module take care of each element of tensor and shift it left or right according the status of that element. In this case, some elements of tensor might be shifted to right, some might be shifted to left and some might remain untouched. This is why we call it non-linear inverse mapping since the shift is not in a uniform direction for the whole tensor.
> >
> > Now I see why you call it linear and non-linear for the two mappings, respectively. However, notice that the major operation is shifting. No matter which direction it shifts, it is always linear. Now since you mentioned overflow or underflow, the fixed-point also has stochastic rounding which is also something that could be considered "non-linear" according to your definitions. I think we shall be more careful about giving these names.
> >
> > In addition, since you mentioned the overflow issue which I have not thought about previously. The author should describe the GEMM math as well in the paper. What's the bit-width of the math? (I think it is 8?) Unlike quantization which uses INT32 for accumulation and it is not easy to overflow, how easy it is to overflow in this technique?
> >
> > > Thanks for your comments, we added Table.4 and also its corresponding paragraph in lines 297-302:
> >
> > Although I like the new integer arithmetic, I would like to see the the code to reproduce this work. Otherwise, it is very difficult to attract the community to support the new technique.
> >
> > > We have performed some experiment on 4 and 5-bits integer numbers, 5-bit integer has around 7% accuracy drop on CIFAR 10 while 4-bit sometimes diverges.
> >
> > I was requesting to perform a comparison to the SOTA just like what happened in Table 4.
> >
> > > We have performed some experiment on 4 and 5-bits integer numbers, 5-bit integer has around 7% accuracy drop on CIFAR 10 while 4-bit sometimes diverges.
> >
> > Please add these to the main text. It would not put the paper in a disadvantageous place. Rather, the community could see the pros and cons more systematically and probably improve the technique in the future.
> >
> > > The loss values of integer and floating-point training are close only if the effect of integer representation mapping is negligible. This is reflected in  which measures variations of fixed point gradient noise. As shown in Figure 3.c of our manuscript, The trajectory of loss for integer and floating-point training are very close. This confirms that  is negligible for this experiment.
> >
> > Variations of fixed point gradient noise, if I understand it correctly, is determined not only by the bit-width but also by the gradient estimator. Probably most of the variations is determined by the gradient estimator. The authors really have to work on this hard to make the readers easy to understand. For example, the authors could say: "Here we argue that by keeping the variance bound M^q relatively small VIA blahblah, we can theoretically achieve the original performance."

---

> > > ### Author Response · Authors · 2022-08-09
> > > **Addressing new comments**
> > >
> > > Thank you for your detailed comments to improve our paper, we have addressed all your comments as follows:
> > >
> > > **Q:  I did read A.1 previously but I had to guess how it would align into the context of the mantissas rounding...**
> > >
> > > Following the example that you mentioned if shifted mantissa is $x= (0.01011001010101010100000)_2$, then $x_1= (0.010110)_2$ and $x_2= (0.010111)_2$. The shifted mantissa $x$ is going to be randomly rounded to either $x_1$ or $x_2$ based on the probability given on equation (13) on line 442 of the manuscript. The realization of stochastic rounding in hardware is added in the **Figure 4** of the modified manuscript. In this figure, the direction of rounding is determined by comparing a random number that is generated on-the-fly with the lower 17-bit of the mantissa.
> > >
> > > [**Click here to see Figure 4**](https://drive.google.com/file/d/1WzbbESp4sYg6BaQ3AsRNm-lN2ahCHVqj/view?usp=sharing)
> > >
> > >
> > > We added an example in the main text **(lines 145-147)**. We have also added the Figure 4 and its explanations to Appendix A.1 due to limited space **(lines 445-447)**
> > >
> > > **Q: Now I see why you call it linear and non-linear for the two mappings, respectively. However, notice that the major operation is shifting. No matter which direction it shifts, it is always linear. Now since you mentioned overflow or underflow, the fixed-point also has stochastic rounding which is also something that could be considered "non-linear" according to your definitions...**
> > >
> > > We do agree that shift is a linear operation, however, alignment which uses shift in different directions for each element is a non-linear operation. Regarding the stochastic rounding, we also agree that it is non-linear, however, one can perform forward pass without stochastic rounding. Stochastic rounding is essential for back-propagation. Moreover, the non-linearity that is introduced by stochastic rounding is negligible since it only affects the unit in the last place (ulp) in the desired precision.
> > >
> > > **Q: In addition, since you mentioned the overflow issue which I have not thought about previously. The author should describe the GEMM math as well in the paper. What's the bit-width of the math? ... how easy it is to overflow in this technique?**
> > >
> > > We clarify that by overflow we did not mean INT32 accumulator, but we meant overflow of 24-bit mantissa which shall be corrected by the alignment module. This phenomenon is also commonplace in the floating-point operation.
> > > In order to clarify the bit-width of math, we added **lines 176-178** in the main text as follows.
> > >
> > > *Note that in our implementation, when the mantissa tensor is in int16 format, multiplication is in int8 format and accumulation is in int32 format.*
> > >
> > > **Q: Although I like the new integer arithmetic, I would like to see the the code to reproduce this work. Otherwise, it is very difficult to attract the community to support the new technique.**
> > >
> > > As mentioned in the checklist of the original paper, the code is proprietary, however we will provide it upon publication.
> > >
> > > **Q: I was requesting to perform a comparison to the SOTA just like what happened in Table 4. ...**
> > >
> > > Thank you for your comments, we have added **Table 5** as well as **lines 312-314** to the main article.
> > >
> > > | **Model** | **Dataset** | int8 | int7| int6 | int5 | int4 |
> > > |---                | ---              | --- | --- | --- | --- | ---
> > > | **ResNet18**  | **CIFAR10**    | 94.8 | 94.7    | 94.47    | 88.5    | Diverges    |
> > >
> > > *Low-bit integer training: Table 5 provides an ablation study of how lowering integer bit-width can affect the training accuracy. Our experiments shows that training has a significant drop of accuracy with int5 and diverges using int4 number formats.*
> > >
> > > Also note that we were not able to do this experiments for all SOTA models because of the limited time that we had for this rebuttal. However, we can infer the extent to which this method works in low-bit regimes from the CIFAR10 experiment in Table 5.
> > >
> > > **Q: Variations of fixed point gradient noise, if I understand it correctly, is determined not only by the bit-width but also by the gradient estimator. Probably most of the variations is determined by the gradient estimator. The authors really have to work on this hard to make the readers easy to understand. For example, the authors could say: "Here we argue that by keeping the variance bound $M^q$ relatively small VIA blahblah, we can theoretically achieve the original performance**
> > >
> > > The variance of the fixed point gradient is only related to the representation and the computation in the desired number format (bit-width). The $M^q$ represents this phenomenon which is not related to the gradient estimator. Thus, $M^q$ only depends on the representation mapping bit-width. The variance of the gradient estimator is reflected in $M$ in Assumption 2 (iii,b), we separated these two constants in order to separate the analysis of their effects. We clarified this on **line 252**.

---

### Official Review · Reviewer_XTiL · 2022-07-10

**Rating:** 6
**Confidence:** 3
**Soundness:** 3 good
**Presentation:** 2 fair
**Contribution:** 2 fair

**Summary:**

This paper proposes a fixed-point based model training pipeline for deep neural networks. It uses linear fixed-point mapping and non-linear inverse mapping to convert the numbers (weights, activations, gradients) during training to fixed-point and back. It conducted theoretical analysis on the noise introduced by the quantization process and the convergence speed with SGD. Experiments are done with several tasks: classification, detection and segmentation. Models include Resnet18, MobileNetV2, ViT-B, DeepLab and FasterRCNN.

**Questions:**

Other issues:
-	On line 33: shouldn’t int8 range be [-127, 128]?
-	Figure 2. On the second block, the right side should be sign2, exponent 2 etc.
-	Table 2&3, the decimal places are not unified.
Overall, my major concern of this work is the over-claim of "int8" pipeline where a lot of operations are not actually int8. And how is the used pipeline different from existing works. More clarifications need to be made. The "int8" bragging and boasting needs to be taken down.

**Limitations:**

Not sure.

**Strengths And Weaknesses:**

Strength:
-	The proposed method achieves pretty impressive results. On ImageNet, the average performance drop for a int8 training pipeline is only around 0.5%. On semantic segmentation, performance drop is around  0.2% and on detection, drop at around 0.3 – 1%.
-	Comprehensive analysis is provided on the impact of int8 training for the SGD convergence, gradient noise and loss landscapes. Although the analysis is rather simplified (e.g. only strong convexity is discussed on the loss landscape part), it still provides interesting insight to the process.
Weaknesses:
-	There are some existing work on using integer arithmetic for training deep models such as (NITI: Training Integer Neural Networks Using Integer-only Arithmetic. TPDS). There is no directly comparison with these methods. The NITI work also uses stochastic rounding which is similar to this work.
-	The authors claim to be using “int8” integer operations for the computation. However, this is misleading since a lot of computation are not “int8”. For example, the multiplication of two int8 numbers/vectors will be using an int16 accumulator. The SGD part is also using int6 arithmetics (line 266). For ViT, the attention computation which contains softmax layers seems not to be using int8 operations (it is not described in the paper).

---

> ### Author Response · Authors · 2022-08-02
> **Response to Reviewer XTiL**
>
> **Q: There are some existing work on using integer arithmetic for training deep models such as (NITI: Training Integer Neural Networks Using Integer-only Arithmetic. TPDS). There is no directly comparison with these methods. The NITI work also uses stochastic rounding which is similar to this work.**
>
> We would like to thank the reviewer XTiL for referring us to the NITI paper. Indeed NITI is an intriguing article, however the approach introduced in NITI shows more than 8\% accuracy drop on ImageNet (Refer to Table 6, last two rows on NITI paper). Moreover, the network architectures used in NITI is limited to AlexNet and VGG which are not state-of-the-art anymore. We have added some lines in the revised manuscript to discuss the above points (lines 103-105).
>
> In regard to stochastic rounding, we have not claimed that our method is the first to use stochastic rounding, please see references [2] and [3] cited on page 2 of our manuscript. Using stochastic rounding is commonplace in the quantization literature, and neither NITI method nor our method is the first to use stochastic rounding.
>
> **Q: The authors claim to be using “int8” integer operations for the computation. However, this is misleading since a lot of computation are not “int8”. For example, the multiplication of two int8 numbers/vectors will be using an int16 accumulator. The SGD part is also using int6 arithmetics (line 266).**
>
> We are certain that nowhere in our manuscript we claimed int8 training. Our article is all about integer arithmetic training where the middle computation such as accumulator is in higher bits as also done in other studies that we are aware of e.g. [1]. For instance, when we multiply two int8 numbers, the result is clearly int16 and consequently for inner product of int8 numbers the accumulator is 32 bits which is standard in scientific computing. Moreover, as you have also noticed, we mentioned in our paper that SGD is in int16 and, therefore, we do not claim int8 training.
>
>
> **Q: For ViT, the attention computation which contains softmax layers seems not to be using int8 operations (it is not described in the paper).**
>
> You are right; aside from ReLU, integer arithmetic for other non-linear functions such as attention has not been considered in our paper. Having said this, integer arithmetic for attention mechanism has recently been explored in other works [2]. The integer arithmetic of their gradients remains however to be thoroughly and systematically studied. A clarification is added to the revised paper in **line 285**.
>
> **Q: On line 33: shouldn’t int8 range be [-127, 128]? - Figure 2. On the second block, the right side should be sign2, exponent 2 etc. - Table 2\&3, the decimal places are not unified.**
>
> Thank you so much for your thorough reading of our manuscript and spotting these typos. We have fixed all these typos in the revised manuscript.
>
> **Q: Overall, my major concern of this work is the over-claim of "int8" pipeline where a lot of operations are not actually int8. And how is the used pipeline different from existing works. More clarifications need to be made. The "int8" bragging and boasting needs to be taken down.**
>
> We did not claim int8 is enough for training deep learning models. The focal point of our work is integer arithmetic and hence the reason for the title. Having said this, we have gone through our manuscripts carefully and re-phrased the sentences that could possibly convey the impression that int8 is enough for training. The main difference of our work compared to others is in method of scaling integer tensors in each layer.
> Our work can be compared to IBM HFP8 [1], where they explore 8-bit floating point values and we explore using int8 values in the training of deep learning models. It is worth noting that in HFP8 paper, the authors have also used higher bit floating-point format such as FP16 for accumulator.
>
> Following your comment we have toned down the abstract. Also please note that the title includes the question mark to hedge this claim.
>
> **References:**
>
> [1] Sun, Xiao, et al. "Hybrid 8-bit floating point (HFP8) training and inference for deep neural networks." Advances in neural information processing systems 32 (2019).
>
> [2] Sari, Eyyüb, Vanessa Courville, and Vahid Partovi Nia. "IRNN: Integer-only recurrent neural network." ICPRAM 11 (2022).

---

> > ### Comment · Reviewer_XTiL · 2022-08-04
> > **Updated comments**
> >
> > Dear authors,
> > Thanks a lot for your clarifications. The revised manuscript is better in toning down the "int8" claims. I recommend to add more explanations on the "middle computation such as accumulator is in higher bits" in the paper.
> > After adding the comparison table to existing methods (Table 4), another concern I have is the performance of this work is not as good as existing work on some model structures (e.g. resnet18). Can the authors provide more explanation on this part? (different settings, quantization schemes etc).

---

> > > ### Author Response · Authors · 2022-08-09
> > > **Updated clarifications**
> > >
> > >
> > > Thank you so much for your detailed comments to improve our paper. We addressed all your comments as follows. If you are satisfied with our answers, please consider increasing your score.
> > >
> > > **Q:  I recommend to add more explanations on the "middle computation such as accumulator is in higher bits" in the paper.**
> > >
> > > Thank you so much for you comments, the clarification is added on **lines 176-178**. The clarification is as follows:
> > >
> > > *Furthermore, note that in our implementation, when the mantissa tensor is in int8 format, multiplication is in int16 format and accumulation is in int32 format.*
> > >
> > > **Q: After adding the comparison table to existing methods (Table 4), another concern I have is the performance of this work is not as good as existing work on some model structures (e.g. resnet18). Can the authors provide more explanation on this part? (different settings, quantization schemes etc).**
> > >
> > > Please note that accuracy of ResNet18 pytorch baseline is **69.75**, our integer implementation has **69.25**  which is 0.5% accuracy drop.
> > > We have three main reasons to justify this:
> > >
> > > *    Our method uses integer batch-norm (forward and back-prop), when other state of the art methods do not.
> > > *    Our method uses integer SGD (int16) while other state of the art methods do not.
> > > *    Other state of the art methods use gradient clipping and hyper-parameter tuning techniques while we just use the original pytorch hyper-parameters.
> > >
> > >
> > > These items are clarified on **lines 304-309** of the updated manuscript.

---

### Official Review · Reviewer_FRGs · 2022-07-13

**Rating:** 6
**Confidence:** 3
**Soundness:** 3 good
**Presentation:** 4 excellent
**Contribution:** 3 good

**Summary:**

The paper proposes a novel integer training method and shows that it is effective to train deep learning models with integer-only arithmetic. More specifically, the proposed method is designed to address and tackle the common challenges observed in integer gradient computation: how to guarantee that the gradients are properly scaled, the numerical error is  unbiased and the training is distribution independent.

**Questions:**

I have listed most of my concerns in previous section and there are several additional questions that might help improve the score.
* Though the authors claim the pipeline is easy to implement and the method is hardware-friendly, it is not that obvious to me as I found the process could be quite complex in practice.
* For Appendix A.2, the bound from equation (18) could become arbitrarily large as $M^q$ and $M^q_v$ are now dependent on the norm of $X$. I am not sure if the bound still makes sense in that case. Maybe there should be some additional assumptions required.

**Limitations:**

It seems there is no negative social impact of the work.

**Strengths And Weaknesses:**

Strengths

* The paper is well-written and the structure is clear.
* Instead of doing quantization, the proposed method directly switches to the number representation of computations and forms a fully integer training pipeline, which is quite novel to me.
* The paper provides a clean theoretical analysis and its motivation is clearly explained in the contribution section.
* The empirical results look good. Compared with baseline, the integer arithmetic performs well.


Weaknesses

* Though the paper is targeted for the experts with quantization computation experience, it would be nice if the authors could elaborate the background and introduce more related work to help the readers to understand the general setup and goal of this area. For a machine learning conference lots of researchers are not that familiar with low-bit integer arithmetic.
* The experiment setup is a little bit too simple. It seems the authors only compare their method with the baselines but not other state-of-art quantization or integer arithmetic training methods. The lack of comparison make the results much less convincible.
* I believe part of the motivation to introduce the integer training pipeline is to reduce the memory complexity and speed up the training. However, I could not see any reports reflecting on these aspects.

---

> ### Author Response · Authors · 2022-08-02
> **Response to Reviewer FRGs**
>
> **Q: Though the paper is targeted for the experts with quantization computation experience, ...**
>
> We would like to thank the reviewer FRGs for this constructive comment. We have added  Appendix A.6 to address your comments. Due to limited space of NeurIPS paper, we have not been able to add this to the main body of the article. However, we properly pointed out the reader in line 108 to refer to this appendix.
>
> **Q: The experiment setup is a little bit too simple. ....**
>
> Thanks for your comments, we have added Table.4 and also its corresponding paragraph in **lines 297-302** where we have compared our proposed method with the state-of-the-art methods.
>
> |**Model**                 | **Dataset**              | **Ours** | **[1]** | **[2]** | **[3]** | **[4]**|
> |---                | ---              | --- | --- | --- | --- | ---
> **MobileNetV2**  | **ImageNet**    | 72.8 | 70.5    | 71.9    | 71.2    | 72.6
> **ResNet18**     | **ImageNet**    | 69.3 | -       | 70.2    | 69.7    | 71.1
> **DeepLab-V1**   | **VOC**         | 74.7 | 69.9    | -       | -       | -
> **Faster R-CNN** | **COCO**        | 37.4 | -       | 37.4    | 34.9    | -
>
> *Comparison with state of the art:  Table 4 provides a comparison between our training method and state of the art across different experiments. There are some important differences between our proposed method and other works: (i) our proposed integer training method uses a fixed-point batch-norm layer where both forward and back propagation is computed using integer arithmetic, (ii) our proposed training method uses an integer-only SGD,  (iii) in our training method, no hyper-parameter is changed while they have changed hyper-parameters or used gradient clipping.*
>
> **Q: I believe part of the motivation to introduce the integer training pipeline is to reduce the memory complexity ...**
>
> The ultimate goal of integer training is to reduce memory footprint and increase the computation throughput. In our paper, we aim to push this research area forward by proposing a new methodology and its theoretical aspects.  The implementation challenges remain to be explored. Our work can be compared to IBM HFP8 [5] paper. In HFP8 [5], the authors introduced a new number format and explored the possibility of performing training and inference with that specific number format without discussing the details of implementation. Later, this number format is used by Nvidia in the recently released hopper architecture.
> Also note that GPUs that support int8 operations are not widespread and we do not have access to them. Moreover, as of the date that we are writing this comment, deep learning frameworks such as Pytorch do not support integer matrix multiplication. We have done this research by using a hardware emulator framework that can exactly emulate the behavior of custom designed arithmetic units on GPU.
>
> **Q: Though the authors claim the pipeline is easy to implement and the method is hardware-friendly ....**
>
> Indeed implementing an integer training framework is not a trivial task. As mentioned in the paper *lines 128-142*, our method consists of *shift* and *round* operations which makes it easier to implement compared to other methods such as [1] and [2]. These methods need to detect the distribution of the data. For instance, author of [6] proposed a new hardware for tackling the problem of distribution detection which is not needed in our proposed method.
>
> **Q: For Appendix A.2, the bound from equation (18) could become arbitrarily large ...**
>
> Although both quantities $M^q$ and $M^q_v$, depends on the norm of $X$, given the ReLU  activation and the batch-norm layer, the norm of $X$ is controlled. As a matter of fact, the training procedure fails to converge even in the floating-point setup if the norm of $X$ increases arbitrarily.
>
> **References:**
>
> [1] Zhang, Xishan, et al. "Fixed-point back-propagation training." Proceedings of the IEEE/CVF Conference on Computer Vision and Pattern Recognition. 2020.
>
> [2] Zhao, Kang, et al. "Distribution adaptive int8 quantization for training cnns." Proceedings of the AAAI Conference on Artificial Intelligence. Vol. 35. No. 4. 2021.
>
> [3] Zhu, Feng, et al. "Towards unified int8 training for convolutional neural network." Proceedings of the IEEE/CVF Conference on Computer Vision and Pattern Recognition. 2020.
>
> [4] Jin, Qing, et al. "F8Net: Fixed-Point 8-bit Only Multiplication for Network Quantization." arXiv preprint arXiv:2202.05239 (2022).
>
> [5] Sun, Xiao, et al. "Hybrid 8-bit floating point (HFP8) training and inference for deep neural networks." Advances in neural information processing systems 32 (2019).
>
> [6] Zhao, Yongwei, et al. "Cambricon-Q: a hybrid architecture for efficient training." 2021 ACM/IEEE 48th Annual International Symposium on Computer Architecture (ISCA). IEEE, 2021.

---

### Meta-Review · Area_Chair_MpUX · 2022-08-28

**Recommendation:** Accept
**Confidence:** Certain

**Metareview:**

This paper proposes methods for using integer arithmetic to train deep learning models. reviewers arrived at a consensus to accept the paper. Concerns do exist on "The title and abstract are misleading to me. There are also floating-point arithmetic in the model, so it is not integer arithmetic only". Hope the author can fix it.

**Award:**

No

---

### Decision · Program_Chairs · 2022-09-14

Accept